

# The impact of core self-evaluation on school adaptation of high school students after their return to school during the COVID-19 pandemic: the parallel mediation of positive and negative coping styles

Qinglin Wang[1,*], Ruirui Duan[2,*], Fulei Han[2], Beibei Huang[2], Wei Wang[2] and Qiulin Wang[2]

[1] College of Medicine, Yangzhou University, Yangzhou, Jiangsu, China
[2] College of Physical Education, Yangzhou University, Yangzhou, Jiangsu, China
[*] These authors contributed equally to this work.

## ABSTRACT

**Background.** To explore the direct effect of core self-evaluation and the indirect effects of positive and negative coping styles on school adaptation of high school students after their return to school during the COVID-19 pandemic.

**Methods.** The Core Self-Evaluation Scale, Simple Coping Style Scale, and School Adaptation Questionnaire were used for the psychometric analysis of 500 high school students (229 males and 271 females) one month after their return to school. The bootstrap method was applied for mediation analysis.

**Results.** A positive correlation was noted between core self-evaluation and school adaptation ($r = 0.56$), and the predictive effect was significant ($\beta = 0.43$). Core self-evaluation positively predicted positive coping styles, which positively predicted school adaptation, while core self-evaluation negatively predicted negative coping styles, which negatively predicted school adaptation. Positive and negative coping styles played a significant mediating role between core self-evaluation and school adaptation. The mediating effect included the indirect effects generated by two pathways: core self-evaluation → positive coping style → school adaptation (95% CI [0.08–0.19]) and core self-evaluation → negative coping style → school adaptation (95% CI [0.03–0.11]).

**Conclusion.** There is a positive association between the core self-evaluation and school adaptation of high school students after their return to school during the COVID-19 pandemic. It may directly or indirectly affect the school adaptation of high school students after their return to school through positive or negative coping styles. After returning to school, educators should guide students to view themselves positively, cultivate healthy core self-evaluation, and enable them to have good school adaptation.

Corresponding author
Qiulin Wang, wangql@yzu.edu.cn

## INTRODUCTION

Since December 2019, the novel coronavirus pneumonia (COVID-19) has spread rapidly worldwide, dramatically changing everyone's lives. In China, in the face of the sudden pandemic, the teaching mode changed from offline to online. During home isolation, students are prone to negative psychological effects. After the pandemic eased, the teaching mode changed again, but the negative psychological impact of the quarantine period remained (*Zhang, Tian & Wu, 2020b*). Therefore, students' mental health after their return to school has become a prime research concern. The mental health burden on students remains high even one to two years after the outbreak (*Windarwati et al., 2022*; *Dhar, Ayittey & Sarkar, 2020*; *El Mzadi et al., 2022*; *Liu et al., 2022*; *Rachel et al., 2022*; *Zheng et al., 2021*), which has a major impact on students' learning and everyday lives after their return to school (*Talevi et al., 2020*). Therefore, it is necessary to ascertain whether life after returning to school affects the school adaptation of high school students. School adaptation results from the interaction between individual students and the school's environment and activities. After returning to school, good school adaptation is manifested by the ability to achieve the educational purpose of the school, successfully complete studies, learn to communicate with others, establish a positive outlook on life, and develop a healthy personality (*Chen, 2006*). Owing to the negative psychological impact of quarantine, students are stressed about returning to school and often experience learning, behavioral, and emotional issues that make it challenging to return to school (*Maria Garcia-Alamino & Tobias, 2021*; *Minkos & Gelbar, 2021*; *Pagerols et al., 2022*; *Scharf, Wiseman & Farah, 2011*; *Zhuo et al., 2021*). Therefore, good school adaptation after high school students' return to school helps them return to school to learn and live, which can reduce or avoid the occurrence of the above problems. Schools need to provide mental health services to students and prepare for the impact of future outbreaks on students' mental health (*Kauffman & Badar, 2022*). To promote a healthy physical and mental development of high school students after their return to school, it is essential to explore the influencing factors and mechanisms of school adaptation after high school students return to school.

A wide range of factors affect students' school adaptation. In addition to external factors such as parent–child relationship, teacher–student relationship, and life events, individual internal factors play a key role in students' school adaptation. Core self-evaluation, as a comprehensive concept of personality, is the most basic assessment of an individual's self-ability and worth, including self-esteem, general self-efficacy, neuroticism, and sources of psychological control. Studies have shown that these four traits are an individual's evaluation of themselves as a whole or some aspects, and a moderate correlation exists between them. At the same time, high core self-evaluation is characterized by high self-esteem, high self-efficacy, high internal control, and low neuroticism. Therefore, core self-evaluation is a relatively persistent and basic evaluation of oneself as an individual (*Smedema, 2014*). According to the theory of conservation of resources, core self-evaluation is an important personality trait when the individual copes with school pressure, resources are sufficient, can meet the external needs, reflected in the school adaptation is good. Conversely, when the individual to cope with the lack of resources of school pressure,

the face of external needs will be unable to cope (*Alarcon, Edwards & Menke, 2011*). The isolation period has been reported to have adverse psychological effects on high school students after their return to school, including negative self-evaluation, inability to quickly adapt to school life, and aggravated difficulty in school adaptation. Studies have a positive correlation between core self-evaluation and school adaptation (*Yu, Chen & Zhang, 2019*). Students with high core self-evaluation tend to exhibit higher academic achievement with better teacher–student and peer relationships and a more harmonious class environment (*Li et al., 2019*; *Ren, Jiang & Ye, 2011*; *Rosopa & Schroeder, 2009*). On the other hand, individuals with low core self-evaluation are more likely to experience academic burnout, internet addiction, and procrastination (*Geng et al., 2018*; *Ma & Li, 2009*; *Shen, Yang & Wang, 2016*). In addition, school adaptation is closely related to self-esteem and self-efficacy, an important part of core self-evaluation, and self-esteem has a significant positive correlation on school adaptation (*Li et al., 2021*; *Wang, 2023*). Furthermore, there is a negative association between core self-evaluation and psychological distress during the COVID-19 pandemic, alleviating suffering during the pandemic (*Shiloh, Peleg & Nudelman, 2022*). Core self-evaluation plays a pivotal role in young people's quality of life and happiness (*Pietsch, Linder & Jansen, 2022*). It can play a regulatory role in mediating negative emotions to enhance the adaptability during the pandemic (*Hameed et al., 2021*). Overall, core self-evaluation is positively correlated with school adaptation. Thus, the first hypothesis of this study is as follows:

## Hypothesis 1: core self-evaluation can positively predict school adaptation

Research has shown that the quality of adaptation is often closely related to how individuals cope. Thus, it needs to be ascertained whether the coping style plays a certain mediating role between core self-evaluation and school adaptation. Coping styles refer to the cognitive and behavioral patterns that individuals use in the face of setbacks and stresses (*Patterson & McCubbin, 1987*). Coping styles reflect an individual's approach to deal with various stressful events and can be divided into positive coping styles and negative coping styles (*Robert, 1986*). Some scholars have pointed out that individuals who practice a positive coping style, actively take effective measures to reduce or eliminate stress results and choose to avoid and endure pressure, while those who follow a negative coping style neglect to solve stress problems (*Peng et al., 2020*). Studies have shown that positive coping styles help middle school students to obtain adaptive development from daily stressful events. In contrast, individuals who often adopt negative coping styles, such as avoidance, are emotionally negative and self-blaming and habitually avoid real stress (*Brand, Laier & Young, 2014*). According to Lazarus' ''evaluation–coping'' theory, individuals with a high level of core self-evaluation tend to take a positive approach to problems (*Lazarus, Folkman & Lazarus, 1966*). Studies have shown a positive correlation between core self-evaluation and positive coping styles during the COVID-19 pandemic (*Zheng et al., 2020*), with positive coping styles mostly adopted by mentally healthy people (*Xia et al., 2021*). Positive coping styles are associated with good mental health (*Wu et al., 2022*), high core self-evaluation, and high self-esteem (*Liu et al., 2016*; *Zhao, Zhang & Ran, 2017*), which can

reduce the impact of psychological symptoms of adolescent and child maltreatment (*Wang et al., 2022*). At the same time, the level of school adaptation is different for different coping styles, and positive coping styles can improve the level of school adaptation. If students respond actively, they can quickly adapt to school life (*Bao, 2012*). As a result, individuals with a higher level of core self-evaluation are better at solving problems in a positive way, actively integrating into school life, and showing good school adaptation (*Shen et al., 2021*). In summary, positive coping styles may play a mediating role between core self-evaluation and school adaptation. Thus, the second hypothesis of this study is as follows:

## Hypothesis 2: positive coping style acts as a mediator between core self-evaluation and school adaptation

According to Lazarus' "evaluation–coping" theory, individuals with low core self-evaluation tend to adopt negative coping styles such as avoiding problems (*Lazarus, Folkman & Lazarus, 1966*) Individuals with low core self-evaluation lack the ability to cope with challenges, treat problems passively and sluggishly, and induce mobile phone dependence by using mobile phones to alleviate negative states (*Liu et al., 2021*). Students with low self-esteem are likelier to choose negative coping styles to solve problems (*Ren, Xi & Ray, 2021*). A negative coping style is the most important related factor of psychological distress during a pandemic (*Li et al., 2022*), and the perceived risk of COVID-19 is positively correlated with psychological distress through a negative coping style (*Chang & Hou, 2022*). A negative coping style is a related factor of back-to-school stress, and appropriate stress management and interventions can help students maintain mental health during the COVID-19 pandemic (*Yuan et al., 2021*). A previous study found that negative coping reduced school adaptation. If students responded negatively, they could not quickly adapt to school life (*Bao, 2012*). Furthermore, coping patterns mediate between sources of psychological control and school adaptation, and choosing a negative coping style can reduce students' school adaptation levels (*Shi, Zhou & Zeng, 2015*). In summary, negative coping patterns may play a mediating role between core self-evaluation and school adaptation. The third hypothesis of this study is as follows:

## Hypothesis 3: negative coping style acts as a mediator between core self-evaluation and school adaptation

A close correlation exists between core self-evaluation, positive coping style, negative coping style, and school adaptation. However, researchers have not studied the mediating roles of positive and negative coping styles on core self-evaluation and school adaptation. High school students' mental health has been severely affected after their return to school during the COVID-19 pandemic. Thus, research on school adaptation after high school students return to school can help improve their mental health after they return to school and adapt to school life. This study analyzed the relevant role of core self-evaluation on school adaptation after high school students' return to school and explored the mediating effect of positive and negative coping styles on school adaptation. The hypothetical model describes the internal mechanisms of the impact of core self-evaluation on school adaptation (Fig. 1).

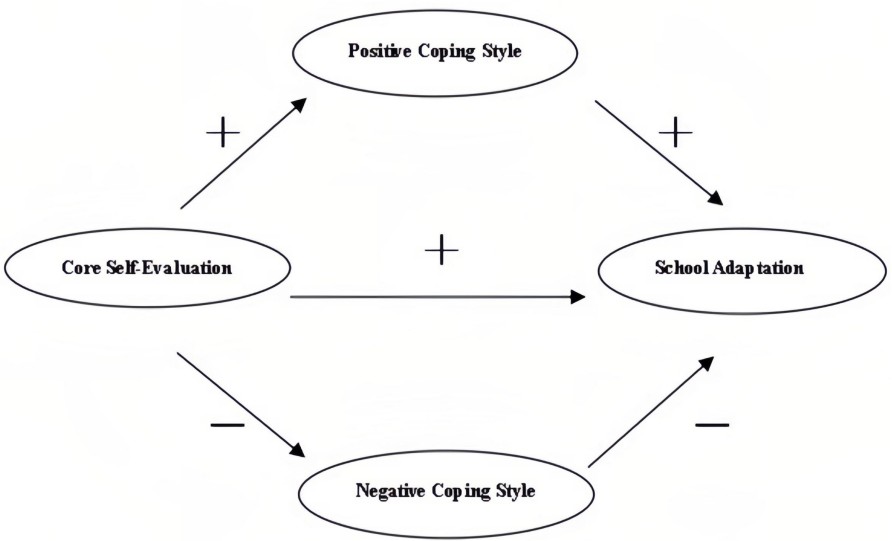

Figure 1  A juxtaposed mediation model of core self-evaluation for school adaptation.

# RESEARCH OBJECTS AND METHODS

## Respondents

A senior high school in Changzhou City, Jiangsu Province, was selected for investigation, and 12 classes of the school were selected by convenient sampling, and four classes from the first to the third year of high school were selected. An informed consent form was issued for the respondents before the investigation to obtain their knowledge and consent to complete the questionnaires. Psychometric paper-and-pencil tests were used, and unified guidance language was used on the spot to guide participants to fill in the questionnaires according to their own real situation, which were collected on the spot. The ethics committee of the medical school of Yangzhou University approved the study (YXYLL-2023-098). Use G*power3.1 software for sample size estimation, use $T$-test for estimation, select correlation analysis, double tailed, set the effect size to 0.8, $\alpha$ to 0.05, power to 0.8, and obtain a sample size of 82. The sample size of this study is consistent with the estimation. Each student simultaneously completed three questionnaires: the Core Self-Evaluation Scale, the Simple Coping Style Scale, and the School Adaptation Questionnaire. A total of 508 people agreed to complete the questionnaire, excluding eight invalid questionnaires, leaving 500 valid samples. The effective rate was 98.4%. The criteria for screening the questionnaire were: (1) complete information; (2) no multi-choice situation for one question; (3) no missed answers.

## Core self-evaluation scale

The modified version of the Core Self-Evaluation Scale, revised by *Du, Zhang & Zhao (2012)*, has 10 questions, such as "I believe I can succeed in life" and "When I fail, I feel very useless." A five-level scoring was used, wherein a score of "1" corresponded to "completely disagreeable," and a score of "5" referred to "fully agree." The higher the

score, the higher the core self-evaluation level, and some items had reversed scoring. In this study, Cronbach's alpha was 0.82, and the confirmatory factor analysis fit index ($\chi^2$/df) was 1.59. Also, the goodness-of-fit index (GFI) was 0.99, the adjusted goodness-of-fit index (AGFI) was 0.98, the corrected goodness-of-fit index (CFI) was 0.99, the incremental goodness-of-fit index (IFI) was 0.99, and the root mean square error of approximation (RMSEA) was 0.03.

### Simple coping style scale

The Simple Coping Style Scale, compiled by *Xie (1998)*, has 20 items including two dimensions: positive coping style and negative coping style. A four-level scoring was used, with 0 and 3 corresponding to "do not take" to "often take," respectively. A higher positive coping style score indicates a higher level of positive coping style, while a lower negative coping style score indicates a higher level of negative coping style. The Cronbach's alpha of the Positive Coping Style Scale in this study was 0.81, and $\chi^2$/df was 2.53. Furthermore, GFI was 0.98, AGFI was 0.96, CFI was 0.98, IFI was 0.98, and RMSEA was 0.06. Cronbach's alpha for the Negative Coping Style Scale was 0.79, and $\chi^2$/df was 2.94. Also, GFI was 0.98, AGFI was 0.96, CFI was 0.98, IFI was 0.98, and RMSEA was 0.06.

### School adaptation questionnaire

The School Adaptation Questionnaire, compiled by *Cui (2008)*, is divided into five parts: routine adaptation, learning adaptation, peer relationship, teacher–student relationship, and school attitude. It has 27 questions, and the higher the score, the higher the level of school adaptation. In this study, Cronbach's alpha of the questionnaire was 0.93, and $\chi^2$/df was 4.06. Furthermore, GFI was 0.88, AGFI was 0.85, CFI was 0.92, IFI was 0.92, and RMSEA was 0.08.

### Statistical methods

Statistical analyses were carried out using social statistical analysis software SPSS 26.0 and AMOS 24.0. The analyses included internal consistency testing of core self-evaluation, coping styles, and school adaptation of high school students after their return to school, and Pearson correlation analysis. AMOS was used for mediation testing and bootstrap analysis. Furthermore, $P < 0.05$ was considered statistically significant.

## RESULTS

### Control and inspection of common method deviations

The Harman univariate test method was used to test whether there existed a common method bias in all project data. The results revealed six factors with eigenvalues greater than 1, the maximum factor variance interpretation rate was 28.87% (<40%), and there were no factors with excessive explanatory power. Thus, there existed no serious common method deviation problem in the data used in this study.

### The relevance of core self-evaluation, coping styles, and school adaptation of high school students after their return to school

Table 1 reports the mean, standard deviation, and correlation of the study variables. Core self-evaluation was positively correlated with school adaptation ($r = 0.56$, $P < 0.001$) and

**Table 1** Analysis of core self-evaluation, positive coping style, negative coping style, and school adaptation.

|  | M ± SD | 1 | 2 | 3 | 4 |
|---|---|---|---|---|---|
| Core self-evaluation | 16.87 ± 4.23 | 1 | | | |
| Positive coping style | 13.16 ± 4.35 | 0.37[**] | 1 | | |
| Negative coping style | 6.00 ± 4.10 | −0.26[**] | 0.05 | 1 | |
| School adaptation | 77.59 ± 12.74 | 0.56[**] | 0.42[**] | −0.31[**] | 1 |

**Notes.**
[*]$P < 0.05$.
[**]$P < 0.01$, same as below.

positive coping styles ($r = 0.37$, $P < 0.001$) but negatively correlated with negative coping styles ($r = -0.26$, $P < 0.001$). School adaptation was positively correlated with positive coping styles ($r = 0.42$, $P < 0.001$) but negatively correlated with negative coping styles ($r = -0.31$, $P < 0.001$).

### The mediating roles of positive and negative coping styles

The model used in this study includes four latent variables, namely, core self-evaluation, positive coping style, negative coping style, and school adaptation, and the model fits well ($\chi^2/df = 2.47$, GFI = 0.85, AGFI = 0.84, CFI = 0.90, IFI = 0.90, RMSEA = 0.05). From this model (Table 2), it can be seen that the path of core self-evaluation to school adaptation reaches the significance level ($P < 0.001$), and hypothesis 1 is verified. Positive and negative coping styles play an intermediary role between core self-evaluation and school adaptation, and this intermediary role includes two paths: the intermediary role of positive coping style and the intermediary role of negative coping style. The deviation-corrected nonparametric percentage bootstrap test was used to repeat the sampling 5,000 times, and the mediation test and the confidence interval were estimated. If the confidence interval did not contain 0, the indirect effect was significant. The results show that positive coping styles (95% CI [0.08–0.19]) and negative coping styles (95% CI: [0.03–0.11]) play a juxtaposed mediating role in core self-evaluation and school adaptation, and hypotheses 2 and 3 were validated (Fig. 2, Table 3). In addition, the effect value of the core self-evaluation on the direct path of school adaptation was 0.43, and the effect values of the indirect path of positive coping styles and negative coping styles were 0.13 and 0.06, respectively. The direct effect accounted for 69.1% of the total effect, and the indirect effect of positive coping styles and negative coping styles accounted for 20.7% and 10.2% of the total effect, respectively.

## DISCUSSION

### The direct effect of core self-evaluation on school adaptation after high school students return to school

The results reveal a positive correlation between core self-evaluation and school adaptation ($r = 0.56$, $\beta = 0.43$). The higher the core self-evaluation level, the better the school adaptation after high school students return to school. The results verify hypothesis 1, and the direct effect of core self-evaluation on school adaptation accounts for 69.1% of the total effect, indicating that core self-evaluation is a key role of school adaptation after high school

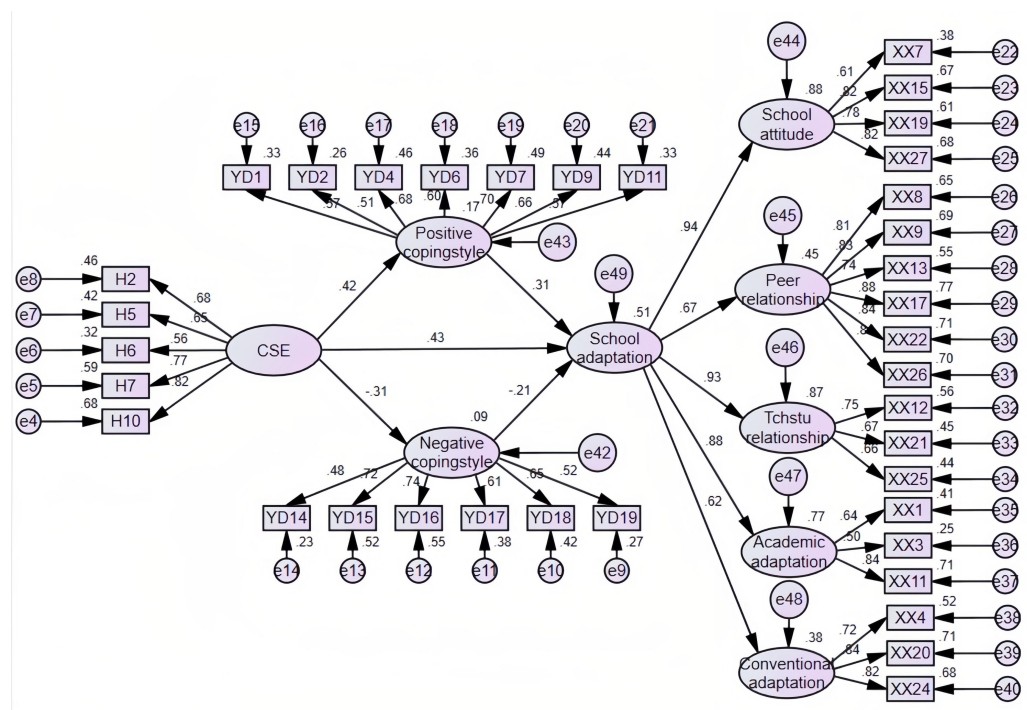

**Figure 2** The parallel mediation model of negative mood state and positive mood state.

**Table 2 Analysis of core self-evaluation, positive coping style, negative coping style, and school adaptation paths.**

|  | SC | SE | CR | *P*-value |
|---|---|---|---|---|
| Core self-evaluation → Positive coping style | 0.42 | 0.03 | 7.01 | *** |
| Core self-evaluation → Negative coping style | −0.31 | 0.04 | −5.28 | *** |
| Positive coping style → School adaptation | 0.31 | 0.06 | 5.71 | *** |
| Negative coping style → School adaptation | −0.21 | 0.05 | −4.26 | *** |
| Core self-evaluation → School adaptation | 0.43 | 0.04 | 7.58 | *** |

**Notes.**
*** $P < 0.001$.

students return to school. In other words, the higher an individual's core self-evaluation level, the higher their school adaptation level. The lower an individual's core self-evaluation level, the lower their school adaptation level. The main function of core self-evaluation is to evaluate self-worth. Major epidemics often occur suddenly, leading to strong emotional reactions in a short period of time, which may damage physical and mental health and lead to poor school adaptation.Previous studies have a positive correlation between core self-evaluation and school adaptation (*Yu, Chen & Zhang, 2019*). Studies have shown that high school students with high core self-esteem have more positive evaluations of themselves, appear more confident in coping with problems in study and life, and are more willing to interact with others and have more harmonious relationships with others (*Chen et al., 2022*; *Dong et al., 2022*; *Li et al., 2019*; *Yang, Li & He, 2021*), which also makes high

**Table 3  Core self-evaluation, positive coping style, negative coping style, school adaptation.**

| Path | Effect value | Effect size | SE | Bias-corrected 95% CI | |
|---|---|---|---|---|---|
| | | | | **Lower** | **Upper** |
| Coreself evaluation → Positive coping style → School adaptation | 0.13 | 20.70% | 0.03 | 0.08 | 0.19 |
| Core self-evaluation → Negative coping style → School adaptation | 0.06 | 10.20% | 0.02 | 0.03 | 0.11 |
| Core self-evaluation → School adaptation | 0.43 | 69.10% | 0.06 | 0.31 | 0.55 |
| Total effect | 0.63 | 100% | 0.04 | 0.53 | 0.71 |

**Notes.**
CI, confidence interval; SE, standard error.

school students more confident to cope with the challenges of learning after returning to school (*Mulyadi, Rahardjo & Basuki, 2016*). In the present study, the high score of core self-evaluation among high school students indicates that the high school students participating in this study maintained good core self-evaluation during the normalization process of COVID-19 pandemic prevention and control. Therefore, educators should properly guide students to view themselves positively, examine themselves with an optimistic attitude, improve their self-confidence, pay attention to students' physical and mental health (*Canbay et al., 2016*; *Şahpolat, Adıgüzel & Arı, 2018*), and help students cultivate healthy core self-evaluation, so that students can have a good school adaptation.

## The mediating roles of positive and negative coping styles

In this study, positive and negative coping styles were used as the mediating variables in the juxtaposed mediation model of core self-evaluation influencing school adaptation after high school students return to school. The effect of mediation shows that the mediation variables are of great significance in explaining the influence of core self-evaluation on school adaptation after high school students return to school. The results of this study indicate that positive and negative coping styles play a mediating role between core self-evaluation and high school students' school adaptation. This discovery provides indirect evidence for the relationship model between school adaptation and coping styles. This model asserts that coping styles are an important protective factor affecting school adaptation. Choosing a positive coping style will improve students' school adaptation level, while choosing a negative coping style will reduce students' school adaptation level (*Amai & Hojo, 2022*; *Shi, Zhou & Zeng, 2015*; *Zhang et al., 2020a*). As a result, individuals with a higher level of core self-evaluation are better at solving problems in a positive way, actively integrating into school life, and showing good school adaptation (*Shen et al., 2021*). Therefore, this study presumes that high school students with high core self-ratings are more likely to establish more stable and good peer relationships after returning to school. Furthermore, they adopt positive coping styles in the face of interpersonal communication and peer communication to adapt to life after returning to school. Such students adopt active and effective learning strategies and complete learning tasks in a timely manner. On the other hand, high school students with low core self-evaluation return to school

and adopt a passive coping style to avoid communication with their peers and fail to adapt well and quickly to life after returning to school. Such students form negative and passive attitudes, and bad emotions such as anxiety and depression and delay learning. Interestingly, in the mediation effect of the structural equation model, the two intermediary paths of positive and negative coping styles are not completely equivalent. The mediation effect of the "core self-evaluation → positive coping style → school adaptation" path accounted for 67% of the overall mediation effect, while the mediation effect of the "core self-evaluation → negative coping style → school adaptation" path accounted for only 33%. Simply put, the core self-evaluation affected school adaptation more through positive coping style styles. This also enlightens educators to guide high school students to cultivate a healthy core self-evaluation after returning to school so that students can have a good school adaptation through a positive coping style.

## Educational implications

Owing to the ongoing changes in the pandemic, high school students are at greater risk of psychological problems when they return to school (*Jardon & Choi, 2022*; *Lee, Jeong & Kim, 2021*; *Ro, Rodriguez & Enriquez, 2021*). In this case, it is especially important to cultivate school adaptation after high school students return to school. In this study, we found that core self-evaluation may play a key role in school adaptation after high school students returned to school, which can be achieved through the mediating role of coping styles. Therefore, we should pay attention to the core self-evaluation, coping style, and school adaptation and provide psychological intervention when necessary to help high school students establish healthy core self-evaluation, better adapt to school life interrupted by the sudden pandemic through positive coping styles, and take positive measures to promote the core self-evaluation, coping styles, and school adaptation of high school students after returning to school in the face of various pressures and problems after returning to school.

## Shortcomings of this study and future prospects

This study has a few limitations. First, this study uses a cross-sectional approach to explore the relationship between various variables, and future longitudinal studies should be used to further analyze the relationship between core self-evaluation, coping styles, and school adaptation, providing empirical research support for exploring the causal relationship between variables and exploring whether it can promote the growth of these three indicators after the return of high school students to school through sports intervention in the future. Second, this study only considers the mediating effect of coping styles between core self-evaluation and school adaptation after high school students return to school. However, in reality, there may be other intermediary variables, such as self-awareness, psychological resilience, and interpersonal skills, which need to be further studied. Third, owing to the restrictions imposed because of the COVID-19 pandemic, the sample size of this study is relatively small, and it should be expanded to supplement and interpret the results of this study.Fourth, due to the impact of the COVID-19 pandemic, the participants were students from only one geographic region. In the future, including participants from multiple regions will help assess the impact of the severity of the epidemic in each region

on students' school adaptation. Fifth, the sample of this study is limited to the group of high school students who returned to school one month after the COVID-19 pandemic. Sixth, convenient sample acquisition has high degree of randomness and selection bias. Seventh, in self-reported data studies, the likelihood of recall bias increases. Finally, this study assessed only high school students. Future studies should include younger and older students to test the findings of this study.

# CONCLUSION

There is a positive association between the core self-evaluation and school adaptation of high school students after their return to school during the COVID-19 pandemic. It may directly or indirectly affect the school adaptation of high school students after their return to school through positive or negative coping styles. After returning to school, educators should guide students to view themselves positively, cultivate healthy core self-evaluation, and enable them to have good school adaptation.

## Contribution to the field

We used structural equation models to analyze the coping styles of 500 returning high school students experiencing the COVID-19 pandemic in China in questionnaires designed to assess core self-evaluation, positive coping styles, negative coping styles, and school adaptation. Our primary results suggest that core self-assessment directly affects school adaptation among returning high school students and indirectly influences their school adaptation through the mediating role of positive and negative coping styles. The positive coping style has a greater impact on school adaptation than the negative coping style, a result consistent with previous research. We believe that the results of this study clarify the internal psychological mechanism of core self-evaluation and school adaptation of returning high school students in the process of pandemic prevention and control and provide suggestions and guidance for returning high school students to adapt to school life after isolation.

# ACKNOWLEDGEMENTS

The authors would like to thank the patients, who contributed in conducting the present study.

## Funding

The authors received no funding for this work.

## Competing Interests

The authors declare there are no competing interests.

## Author Contributions

- Qinglin Wang conceived and designed the experiments, performed the experiments, prepared figures and/or tables, authored or reviewed drafts of the article, and approved the final draft.
- Ruirui Duan conceived and designed the experiments, performed the experiments, prepared figures and/or tables, and approved the final draft.
- Fulei Han performed the experiments, analyzed the data, prepared figures and/or tables, and approved the final draft.
- Beibei Huang performed the experiments, analyzed the data, prepared figures and/or tables, and approved the final draft.
- Wei Wang performed the experiments, analyzed the data, prepared figures and/or tables, and approved the final draft.
- Qiulin Wang conceived and designed the experiments, performed the experiments, authored or reviewed drafts of the article, and approved the final draft.

## Human Ethics

The following information was supplied relating to ethical approvals (i.e., approving body and any reference numbers):

The medical school of Yangzhou University approved the study (YXYLL-2023-098).

## Ethics

The following information was supplied relating to ethical approvals (i.e., approving body and any reference numbers):

The medical school of Yangzhou University approved the study (YXYLL-2023-098).

## Data Availability

The raw data are available in the Supplemental Files.

## Supplemental Information

Supplemental information for this article can be found online at http://dx.doi.org/10.7717/peerj.15871#supplemental-information.

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
