# Peer review of "The impact of core self-evaluation on school adaptation of high school students after their return to school during the COVID-19 pandemic: the parallel mediation of positive and negative coping styles"

_PeerJ, doi:10.7717/peerj.15871_

## Round 0.1 · original submission · Minor Revisions

I have now received the reviewers' comments on your manuscript. They have suggested some minor revisions to your manuscript. Therefore, I invite you to respond to the reviewers' comments and revise your manuscript.

·

Basic reporting

Dear Editor,
I really appreciate the opportunity to review the manuscript peerj-83721 entitled:
"The impact of core self-evaluation on school adaptation of high school students after their return to school during the COVID-19 pandemic: the parallel mediation of positive and negative coping styles"

Experimental design

I commend the authors for describing this critical and timely issue. The paper is interesting and well-written; however, I would like to highlight some issues that merit revision:

Validity of the findings

It is not particularly clear from the manuscript whether the students evaluated had access to some form of psychotherapy or counseling either in-person or possibly at a distance through telemedicine. Since this is a particularly important factor an application of it could be a confounding factor. I ask the authors to add a brief passage on this aspect, or if the data is not available to add it to the limitations

Additional comments

None

Reviewer 2 ·

Basic reporting

...

Experimental design

...

Validity of the findings

....

Additional comments

Manuscript title: The impact of core self-evaluation on school adaptation of high school students after their return to school during the COVID-19 pandemic: the parallel mediation of positive and negative coping styles
This is important clinical study.
Some other comments to consider:
1.Abstract should be written correctly,
2.The discussion chapter is not well written and looked deeply in to the literature and cited to international research. In overall the references list is rather old, latest from 2018 (rather less), consider updating?
3.It may be important that you should cite and discuss below articles;
-Sahpolat M, Adiguzel M, Ari M. Focusing on physical symptoms and psychological trauma of patients with bruxism. Bakırköy Tıp Dergisi 2018;14:283-8.
-Tambag H, Sahpolat M. Alexithymia and Anger in Patients with Bruxism.International Journal of Caring Sciences. 2021;14(1):507-514.
-Canbay Ö, Doğru E, Katayıfçı N. Investigation of obesity frequency and eating habits in a university hospital professionals. Medical Journal of Bakırköy 2016; 12: 129-135.
4.The study has a lot of limitations. Authors have written rather little about it. You should write more limitations of this study.
5. You must arrange the references have not been arranged according to journal guideliness.
Sincerely

·

Basic reporting

This is a very well-written manuscript. The Introduction section is well structured describing the background and the gap in the literature. The specific problem being investigated is clearly articulated. The entire manuscript is professionally structured, raw data is provided, and appropriate statistical methods have been used snd described.

Experimental design

The work is with in the scope of the journal. The research questions are well-described ( effect of core self evaluation on school return and the mediating effect of coping styles). Appropriate statistical equation modelling is used. Enough details is provided for the study to be repeated. Limitations are acknowledged and well-described.

Validity of the findings

The authors have provided the raw data and as stated earlier, enough detail about the methods, analysis and findings. The interpretations are valid an justified. Conclusions are reasonable and limited only to study findings.

Additional comments

Very well-written. Only minor grammatical changes are required that could be easily addressed through help with software or review by a language expert or editor.

---

## Round 0.2 · Minor Revisions

Thank you for the update. However, there are still concerns that prevent me from accepting the revised paper:

GENERAL REVIEW
As you mentioned in the limitations section, the cross-sectional research cannot explore the causal relationship between study variables. Therefore, I suggest to replace "prediction" with "association" or "relationship" in all parts of the article.

ABSTRACT
The abstract is not acceptable in its current form. In the abstract, there is no mention of conclusions.

METHODS
In methods section, did you perform power analysis? Please describe the sample size, power, and precision.

It is still unclear to me whether the study data were normal or not.

RESULTS
Please round the numbers (except p-valus) to two decimal places.

DISCUSSION
Convenience samples never result in a statistically balanced selection of the population. This leads to selection bias. So, It is necessary to mention this limitation of the study. Another limitation of the study is the increased chance of recall bias in study using self-reported data.

·

Basic reporting

Dear Editor,
I really appreciate the opportunity to review the manuscript peerj-83721 entitled:
"The impact of core self-evaluation on school adaptation of high school students after their return to school during the COVID-19 pandemic: the parallel mediation of positive and negative coping styles"

Experimental design

The paper is very interesting and well-written, methodologically unexceptionable, and the new implementations provide a valid contribution to the work. Every requested correction has been done, and the manuscript is now suitable for publication

Validity of the findings

No issues detected

Additional comments

No issues detected

---

## Author Rebuttal · Round 0.2

Dear Editors and Reviewers,

Thanks very much for taking your time to review this manuscript. We really appreciate all your generous comments and suggestions! According to your advice, we amended the relevant part in manuscript. All of your questions were answered one by one. Revised parts have been marked with different color (red). Please find my revisions in the re-submitted files. Other changes in the article were found and corrected by experts after they were polished.

**Reviewer#1**

1.It is not particularly clear from the manuscript whether the students evaluated had access to some form of psychotherapy or counseling either in-person or possibly at a distance through telemedicine. Since this is a particularly important factor an application of it could be a confounding factor. I ask the authors to add a brief passage on this aspect, or if the data is not available to add it to the limitations.

1:Line 339 to 340:Thank you very much for your suggestion. We have added restrictions.

**Reviewer#2**

1.Abstract should be written correctly.

1:Line 20 to 27:Thank you very much for your suggestion.We have adjusted the format.

2.The discussion chapter is not well written and looked deeply in to the literature and cited to international research. In overall the references list is rather old, latest from 2018 (rather less), consider updating?

2:Line 259 to 276;Line 284 to 292:Thank you very much for your suggestion.We have updated the literature.

3.It may be important that you should cite and discuss below articles;
-Sahpolat M, Adiguzel M, Ari M. Focusing on physical symptoms and psychological trauma of patients with bruxism. Bakırköy Tıp Dergisi 2018;14:283-8.
-Tambag H, Sahpolat M. Alexithymia and Anger in Patients with Bruxism.International Journal of Caring Sciences. 2021;14(1):507-514.
-Canbay Ö, Doğru E, Katayıfçı N. Investigation of obesity frequency and eating habits in a university hospital professionals. Medical Journal of Bakırköy 2016; 12: 129-135.

3:Line 268 to 269;Line 290;Line 274 to 276:Thank you very much for your

suggestion.We have discussed and cited some literature.

4.The study has a lot of limitations. Authors have written rather little about it. You should write more limitations of this study.

4:Line 336 to 342: Thank you very much for your suggestion.We have added some restrictions.

5.You must arrange the references have not been arranged according to journal guideliness.

5:Line 382 to 561: Thank you very much for your suggestion.We have adjusted the literature format.

**Reviewer#3**

1.Only minor grammatical changes are required that could be easily addressed through help with software or review by a language expert or editor.

1:Thank you very much for your suggestion.We have already adjusted.

---

## Round 0.3 · accepted · Accept

Many thanks for addressing all the issues. However, there are still grammatical mistakes in parts of the article, which I hope will be fixed in the galley proof of the article.